# Predicting Optical Water Quality Indicators from Remote Sensing Using Machine Learning Algorithms in Tropical Highlands of Ethiopia

**Elias S. Leggesse** [1], **Fasikaw A. Zimale** [1], **Dagnenet Sultan** [1], **Temesgen Enku** [1], **Raghavan Srinivasan** [2] and **Seifu A. Tilahun** [1,3,*]

[1] Faculty of Civil and Water Resources Engineering, Bahir Dar Institute of Technology, Bahir Dar University, Bahir Dar H9FX+Q62, Ethiopia; 2003sime@gmail.com (E.S.L.); fasikaw@gmail.com (F.A.Z.); dags120120@gmail.com (D.S.); temesgenku@gmail.com (T.E.)
[2] Temple Research and Extension Center, Texas A&M AgriLife Extension, Texas A&M AgriLife Research, Temple, TX 75684, USA; r-srinivasan@tamu.edu
[3] International Water Management Institute, Accra PMB CT 112, Ghana
[*] Correspondence: s.tilahun@cgiar.org or satadm86@gmail.com

**Abstract:** Water quality degradation of freshwater bodies is a concern worldwide, particularly in Africa, where data are scarce and standard water quality monitoring is expensive. This study explored the use of remote sensing imagery and machine learning (ML) algorithms as an alternative to standard field measuring for monitoring water quality in large and remote areas constrained by logistics and finance. Six machine learning (ML) algorithms integrated with Landsat 8 imagery were evaluated for their accuracy in predicting three optically active water quality indicators observed monthly in the period from August 2016 to April 2022: turbidity (TUR), total dissolved solids (TDS) and Chlorophyll a (Chl-a). The six ML algorithms studied were the artificial neural network (ANN), support vector machine regression (SVM), random forest regression (RF), XGBoost regression (XGB), AdaBoost regression (AB), and gradient boosting regression (GB) algorithms. XGB performed best at predicting Chl-a, with an $R^2$ of 0.78, Nash–Sutcliffe efficiency (NSE) of 0.78, mean absolute relative error (MARE) of 0.082 and root mean squared error (RMSE) of 9.79 µg/L. RF performed best at predicting TDS (with an $R^2$ of 0.79, NSE of 0.80, MARE of 0.082, and RMSE of 12.30 mg/L) and TUR (with an $R^2$ of 0.80, NSE of 0.81, and MARE of 0.072 and RMSE of 7.82 NTU). The main challenges were data size, sampling frequency, and sampling resolution. To overcome the data limitation, we used a K-fold cross validation technique that could obtain the most out of the limited data to build a robust model. Furthermore, we also employed stratified sampling techniques to improve the ML modeling for turbidity. Thus, this study shows the possibility of monitoring water quality in large freshwater bodies with limited observed data using remote sensing integrated with ML algorithms, potentially enhancing decision making.

**Keywords:** water quality; Landsat; machine learning; Lake Tana

## 1. Introduction

As populations increase and agriculture intensifies, eutrophication of freshwater bodies around the globe, particularly in Africa, is becoming a major challenge [1]. Water quality monitoring is essential to identify water contamination sources and implement best management practices for healthy ecosystems [2]. It is standard practice to monitor various water quality parameters through field observation, i.e., collecting water samples from various spatial locations at various temporal resolutions. However, data collection in the field is labor-intensive, time-consuming, and expensive. Moreover, data obtained by field observation are intermittent in space and time, and it is almost impossible to predict water quality trends in large waterbodies from such data [3]. Consequently, monitoring water

quality using a combination of remote sensing and machine learning (ML) is seen as an increasingly attractive alternative to standard field monitoring.

Remote sensing techniques enable monitoring water quality issues more effectively and efficiently in large-scale regions and water bodies. They are widely used for water quality assessment in the contemporary world [2,4]. Kallio [4] demonstrated the advantages of applying remote sensing, including its synoptic view of an entire water body and its ability to provide historical water quality data at larger spatial and various temporal scales. Kibena et al. [5] identified various techniques for retrieving water quality parameters from remote sensing imagery, including empirical, semi-empirical, and analytical methods. Recently, the potential application of ML algorithms to monitor water quality parameters in inland water from various sensors has been tested by different researchers [6–10] and has shown a promising result. The use of ML in retrieving water quality parameters from remote sensing images is still in its early stage and several challenges exist that affect its accuracy.

The major concern Is the optical complexity of waterbodies, which affects the accuracy of ML algorithms [9,10]. Another major setback is the lack of continuous long-term spatiotemporal data representing the underlying dynamic of the water quality parameter in the waterbody system to be used for model training and testing [6,7]. The frequency, depth, and spatial resolution of sampling are also important issues affecting ML models' accuracy [8,9]. Overall, integrating ML with remote sensing would potentially provide additional benefits in retrieving water quality parameters [11].

Chen et al. [7] compared the water quality prediction performance of 10 machine learning models (seven traditional and three ensemble models) using big data from the major rivers and lakes. Their results indicated decision trees (DT), random forest (RF) and deep cascade forest (DCF) could be prioritized for future water quality monitoring and providing timely water quality information. Bui et al. [9] introduced novel hybrid data-mining algorithms (combinations of standalones with bagging (BA), CV parameter selection (CVPS) and randomizable filtered classification (RFC)) to perform water quality predictions. They found that hybrid algorithms improved several standalone models' prediction power, but not all. An important step in machine learning is identifying input variables that predict the selected water quality parameter. Sudher et al. [11] proposed an analytical approach to identify the appropriate combination of input variables (remote sensing band data). This approach could significantly reduce the effort and computational time required to develop a water quality model.

The most commonly used machine learning models for optical water quality parameters are RF, SVR, and ANN. Kim et al. [12], for example, attempted to estimate two optical water quality indicators, chlorophyll-a (chl-a) and suspended particulate matter (SPM) concentrations, in coastal environments using random forest and support vector regression (SVR) and showed that SVR outperformed the other two machine learning approaches. Park et al. [13] used an artificial neural network (ANN) and a support vector machine (SVM) to predict Chl-a concentration. Their study revealed that the two models reproduced the temporal variation of Chl-a well, based on the weekly input variables. In particular, the SVM model performed better than the ANN model, displaying higher prediction accuracy in the validation step. All these ML prediction attempts were tested in a temperate climate.

Water quality prediction using ML in the monsoonal climates of Ethiopia has not been widely investigated. Recent research has focused more on retrieving water quality parameters via remote sensing, identifying the sources of water quality parameters, mapping the distribution of certain water quality parameters, and trying to understand the transport mechanisms of certain water quality parameters of Lake Tana, the largest freshwater body in Ethiopia. Moges et al. [14] retrieved current and previous trends in the water quality parameters Chlorophyll a (Chl-a), turbidity (TUR), Secchi disc transparency depth (STD), and dissolved phosphorus concentration (DPC) from remote sensing using the regression method. Dersseh et al. [3,15] assessed spatial distributions of total phosphorus (TP), total nitrogen (TN), water surface temperature (T), total dissolved solids (TDS), and pH using geostatistical analysis in ArcGIS. Additionally, Goshu et al. [16] assessed the seasonality

and sources of dissolved inorganic nitrogen (DIN) inputs into Lake Tana of the upper Blue Nile River basin. Some of these studies [3,14–16] indicate that the water quality of Lake Tana is degrading with time, suggesting that immediate implementation of best management practices is necessary, as well as continuous water quality monitoring to evaluate the effectiveness of such practices.

Continuous field monitoring of various water quality parameters is impractical and costly in large lakes such as Lake Tana. Moges et al. [14] showed that the retrieval algorithm based on a non-linear regression model was unreliable, particularly for Chl-a monitoring. However, prior studies in the upper Blue Nile River basin have found the use of remote sensing and ML to be effective in stream flow forecasting [17], precipitation evaporation index [18], and soil moisture monitoring [19]. It is, therefore, worth investigating the potential applications of this approach for water quality monitoring. This approach would likely prove more practical and effective than continuous field monitoring of water quality in large lakes and similar water bodies.

Chl-a, TDS, and TUR are the most common water quality indicators for water bodies with agricultural upland watersheds. They could potentially be retrieved via remote sensing due to their optical properties [20]. Chl-a is a major optical water quality parameter because it links nutrient concentration (particularly phosphorus) and algal production. Another optical property of water, TUR, scatters and absorbs light rather than transmitting it. Total dissolved and suspended solids are responsible for most of the scattering, whereas absorption is controlled by Chl-a and colored dissolved or particulate matter [1]. TUR and total suspended solids are important variables in many studies due to their linkage with incoming sunlight, affecting photosynthesis for the growth of algae and plankton in water bodies [1].

Thus, the main objective of this study was to evaluate the performance of various ML models in monitoring TDS, TUR, and Chl-a when combined with LANDSAT-derived imagery in a data-scarce area of Ethiopia. The study selected six ML algorithms for monitoring these parameters in Lake Tana: the artificial neural network (ANN), support vector machine regression (SVM), random forest regression (RF), XGBoost regression (XGB), AdaBoost regression (AB), and gradient boosting regression (GB). Unlike previous studies, here we presented a large set of features as inputs to the ML models, and the models would select the best features that give the model the highest performance based on its internal structure and its ability to extract useful information from the features. Furthermore, to overcome the data limitation, we used a K-fold cross validation technique that could obtain the most out of the limited data to build robust models. We also employed a stratified sampling technique to improve the ML modeling for the most data-limited parameter. Hence, even with limited data, the study demonstrated that a reliable model could be built to help monitor important water quality variables through researching various available options in the ML modeling.

## 2. Materials and Methods

### 2.1. Study Area

Lake Tana is Ethiopia's largest freshwater lake and the outlet of the Lake Tana sub-basin of the upper Blue Nile (Abbay) River Basin. The lake is located at latitude $12°00'$ N and longitude $37°15'$ E, at an elevation of 1786 m above sea level (Figure 1). Its surface area varies from approximately 3050 km$^2$ during the dry season to approximately 3600 km$^2$ near the end of the rainy season. It runs 68 km east to west and 73 km north to south, with a maximum depth of 14 m and an average depth of 9 m [21]. The United Nations Educational, Scientific and Cultural Organization (UNESCO) has designated the lake as a biosphere reserve [22].

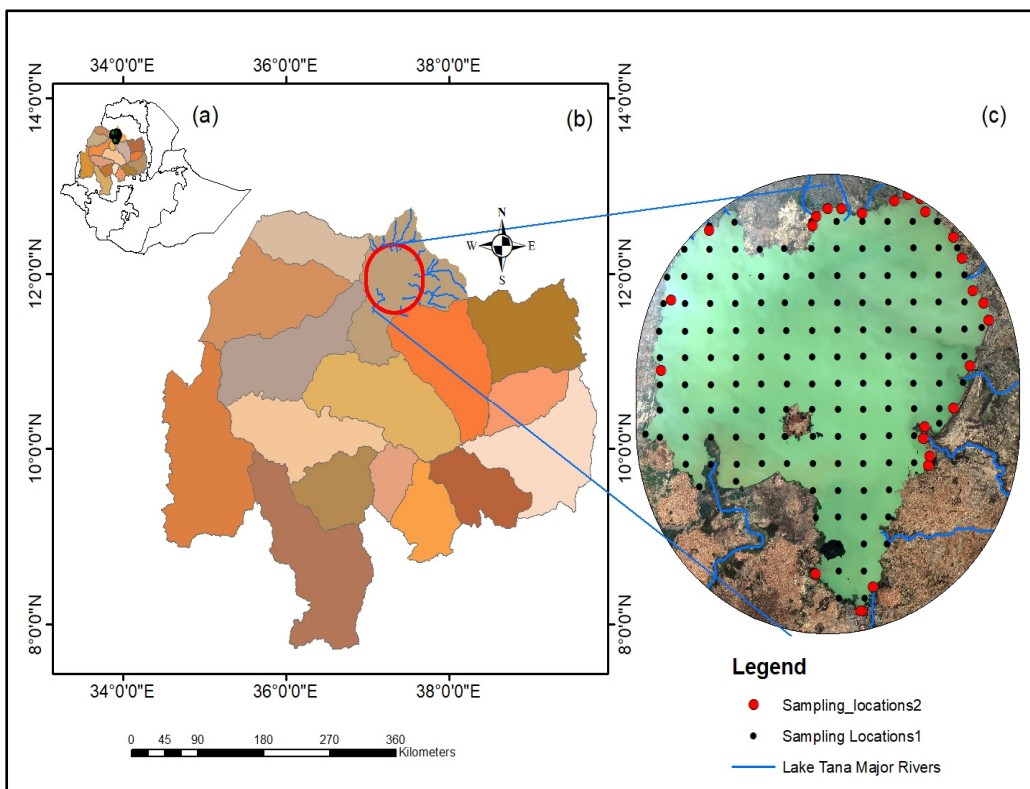

**Figure 1.** Map of Ethiopia (**a**), Blue Nile Basin (**b**), and Lake Tana with water quality sampling point locations (**c**). Black dots represented sampling periods of August 2016, December 2016, March 2017, October 2021, and March 2022. The red dots represented sampling periods of June 2019, July 2019, August 2019, September 2019, December 2019, and March 2020.

The climate of the Lake Tana sub-basin is a warm-temperate tropical highland monsoon climate with large diurnal temperature variation between daytime highs of 30 °C and nighttime lows of 6 °C; a mean temperature of 21.7 °C; seasonal variation of only about 5 °C, and two annual peaks in temperature; one in May/June and one in October/November [21]. Between October/November and May/June, there is a dry season; between July and September, there is a distinct rainy season (kiremt) [23]. The average annual rainfall on the lake varies from 1600 mm in the southern part to 1200 mm in the northern part [24].

The Lake Tana sub-basin has a drainage area of 15,054 km$^2$, of which the lake accounts for 20%. Over 60 rivers and streams drain from the watershed into Lake Tana. The six main tributary rivers are the Gilgel Abay, Gumera, Ribb, Gelda, Megech, and Dirma Rivers. The lake receives the majority of agricultural and urban runoff and domestic waste effluents from the three major cities of Bahir Dar, Gonder, and Debre Tabor [25]. Seventy-five percent of the sub-basin land is agricultural land planted with rainfed teff, maize, and sorghum crops and irrigated crops of onion, tomato, maize, and wheat [26,27].

### 2.2. Water Samples Collection and Laboratory Analysis

A monthly dataset used to conduct this research was acquired from previous research [15], and primary data were collected in 2021 and 2022. The data include three optically active water quality indicators (Table 1): TUR, TDS, and Chl-a. The number of samples collected and analyzed for Chl-a, TDS, and TUR were 931, 796, and 286, respectively. The dataset included data gathered from 143 sampling locations across the lake at a 5 km resolution from the top water surface at a depth of 50 cm in August 2016, December 2016, March 2017, October 2021, and March 2022 (Figure 1: sampling_locations1). A second dataset included data gathered from 27 sampling locations in June 2019, July 2019, August 2019,

September 2019, December 2019, and March 2020 (Figure 1: sampling_locations2). The sampling dates for Chl-a were chosen to represent the main rainy season (July–September), the dry season (December–April), and the pre-rainy season (May–June) to understand how seasonality influences the water quality parameters. The water quality parameters TUR, TDS, and Chl-a were analyzed in the Bahir Dar Institute of Technology water quality laboratory.

**Table 1.** Descriptive statistics of each water quality parameter measured in the field campaign and laboratory analysis. Statistical metrics used were: maximum value (Max), minimum value (Min), mean, standard deviation (SD).

| Water Quality Parameter | Metrics | Aug. 2016 | Dec. 2016 | Mar. 2017 | Dec. 2019 | Jun. 2019 | Jul. 2019 | Aug. 2019 | Mar. 2020 | Oct. 2021 | Apr. 2022 |
|---|---|---|---|---|---|---|---|---|---|---|---|
| N. Sample | | 170 | 170 | 170 | 27 | 27 | 27 | 27 | 27 | 143 | 143 |
| Chl-a (µg/L) | SD | 0.1 | 0.1 | 0.10 | 1.1 | 4.0 | 3.3 | 2.0 | 1.8 | 0.76 | 16.8 |
| | Max | 19.4 | 191.6 | 191.6 | 7.2 | 19.5 | 14.2 | 11.4 | 12.0 | 185.8 | 125.0 |
| | Min | 0.05 | 0.05 | 0.05 | 3.7 | 0.8 | 1.1 | 1.5 | 6.2 | 0.01 | 0.01 |
| | Mean | 2.2 | 17.1 | 20.6 | 5.4 | 4.4 | 5.7 | 6.3 | 8.8 | 6.7 | 8.6 |
| TUR (NTU) | SD | - | - | - | - | - | - | - | - | 19.0 | 19.7 |
| | Max | - | - | - | - | - | - | - | - | 344 | 104.0 |
| | Min | - | - | - | - | - | - | - | - | 0.27 | 5.0 |
| | Mean | - | - | - | - | - | - | - | - | 41.7 | 23.7 |
| TDS (mg L$^{-1}$) | SD | 2.8 | 26.4 | 25.7 | - | - | - | - | - | 34.0 | 5.7 |
| | Max | 113.3 | 107.3 | 107.3 | - | - | - | - | - | 99.0 | 78.0 |
| | Min | 50.7 | 34.7 | 34.7 | - | - | - | - | - | 30.0 | 7.30 |
| | Mean | 90.1 | 87.0 | 99.4 | - | - | - | - | - | 58.6 | 68.2 |

A TDS meter, also known as an electrical conductivity meter, was used to measure TUR and TDS. Chl-a concentrations were measured using the acetone extraction method after sample filtration on a 0.47 m glass fiber filter (Whatman GF/C) utilizing Gellman polycarbonate filtration towers at low-to-moderate vacuum (10–40 cm Hg). Centrifugation at 4000 rpm for 20 min cleared the extracts. Before acidification (750b and 664b) and after acidification (750 and 665 nm), sample and standard absorbance were measured at 750 and 664 nm (750a and 665a). The concentration of Chl-a in the extract was evaluated using a spectrophotometer equipped with a Perkin-Elmer Lambda 35 UV/VIS spectrophotometer with a 1 nm spectral bandwidth and optically matched 4 cm plastic micro-cuvettes, as per the standard method. The descriptive statistics of the three selected optical water quality parameters of Lake Tana are shown in Table 1.

The measured values of Chl-a, TDS, and TUR in the laboratory analysis have had an outlier that prompted us to use an outlier detection and removal techniques. This study used a density-based technique called the local outlier factor (LOF) algorithm because of its simplicity and ease of use, without considering data distribution.

### 2.3. Landsat 8 OLI Image Acquisition and Pre-Processing

Landsat 8 Operational Land Imager (OLI) images were used in this study. NASA successfully launched Landsat 8 on 11 February, 2013. Even though new sensors such as the Landsat 8 OLI lack certain band centers useful for inland water remote sensing, they have improved signal-to-noise ratios, radiometric and temporal resolution, and aerosol-specific bands, making them better equipped to handle the size and complexity of inland waters [28]. The Landsat 8 OLI images were collected from "USGS Landsat 8 Surface Reflectance Tier 1" of the Google Earth Engine (GEE) dataset with 30 m spatial resolution. Tier 1 data were corrected for atmospheric and geometric errors (http://earthexplorer.usgs.gov, accessed 11 December 2021). With the help of GEE, the spectral band reflectance values were extracted at each sampling point for the sampling months of August 2016, December 2016, March 2017, June 2019, July 2019, August 2019, September 2019, December 2019, March 2020, October 2021, and March 2022.

Explanatory variables used to train the ML models were extracted from these Landsat 8 OLI bands and derived spectral indices based on image differentiating (DI), ratio remote sensing index (RI), and different types of normalized remote sensing indices (NDI). As shown in Table S1, in addition to nine bands (B1, B2, B3, B4, B5, B6, B7, B11, B12) of Landsat 8 OLI imagery, the study identified 78 spectral indices, including NDWI of McFeeters [29], NDWI of Rogers and Kearney [30], MNDWI of Xu [31], AWEInsh and AWEIsh of Feyisa et al. [32], FAI (Floating Algae Index) [33], different normalized difference vegetation indices, and several other water vegetation indices as proposed explanatory variables. These variables are listed by index name and a short description of each can be found in Table S1 of the Supplementary Material. Before the features were used directly in the modeling, a MinMax Scaling was applied to rescale the data in a fixed range of 0 and 1. The method subtracts the minimum value in the feature and then divides it by the range.

### 2.4. Model Description and Approach

This study selected six ML models frequently used in water quality assessment and monitoring: RF, XGB, AB, GB, SVM, and ANN. The selected ML models were trained using three optical water quality parameters as target variables: TUR, TDS, and Chl-a. As shown in Figure 2, the explanatory selected feature variables outlined in Section 2.4.1 were used as input, and the models were evaluated using different metrics, as discussed in Section 2.5 below. Model hyperparameters were tuned using the grid search method [34,35] with K-fold cross validation (CV), with the number of folds K set to five, as recommended by other studies [36–38]. Figures S1–S36 in the Supplementary Material showed the optimization results for the ML models from the grid search method. The ML model parameters with the lowest MSE and the selected features were used to construct the ML models.

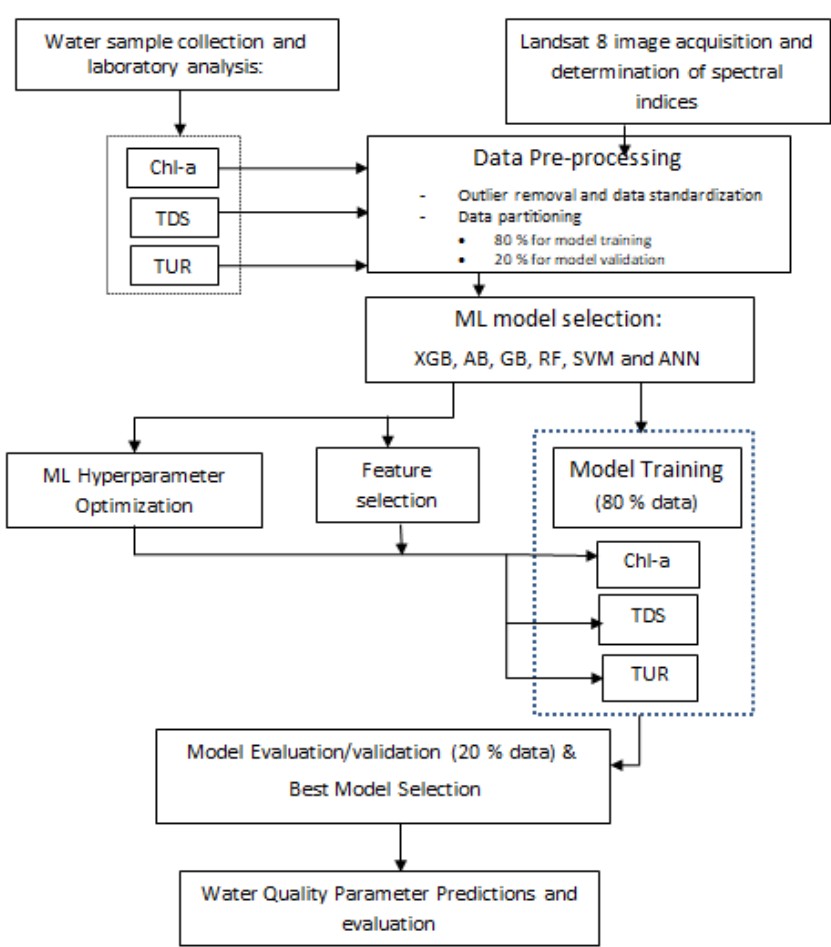

**Figure 2.** The overall study framework used in this study.

The study used a predefined ANN model, as outlined in Section 2.4.6. A wrapper technique called recursive feature elimination with cross-validation (RFECV) was used for feature selection to speed up the learning algorithm, improve predicted accuracy, and make learning results more intelligible [37,38]. Eighty percent of the dataset was used for training and twenty percent for testing. We used the traditional random sampling approach (TRSA) to split data for all water quality parameters. As the prediction with this sampling approach was poor for TUR, we applied an additional approach, a spatiotemporal block partitioning technique (STBPT), to carry out the split [39].

After training and testing the ML models, the best model for each water quality parameter was selected based on its performance metrics and used to predict the monthly spatial distribution and a spatial monthly average of Chl-a for the year 2020, and TUR and TDS for the year 2017. The target years were chosen to coincide with the available studies of the various parameters. These predictions were based on Landsat 8 (OLI) images and the selected features for each model, as extracted on 3065 points at a 1 km resolution and monthly intervals for selected years. The spatial mapping was conducted by interpolating the 3065 predicted water quality parameters using the inverse distance weightage method. The mean monthly spatial average was calculated by averaging the predicted monthly values for these 3065 points. These predictions were used to compare the predictive abilities of ML models with other methods in the literature [40–42] and to assess the water quality of Lake Tana.

### 2.4.1. Adaboost (AB)

AB is a typically boosting type ensemble ML algorithm introduced by Freund [43]. It trains the weak learners and then integrates the trained weak learners to obtain a final model [44]. AB assigns different weights (called "amount of say") to the prediction error rate of the learner, then adjusts the weight of the sample, and finally, accumulates and weights the prediction results of all learners to generate a predicted value.

### 2.4.2. Random Forest (RF)

RF is a tree-based ensemble technique proposed by Breiman [45]. A regression tree is a non-linear regression model where samples are partitioned at each binary tree node based on the value of one selected input feature. It is a classification model that uses multiple base models independently, typically decision trees, on a given subset of data, and makes decisions based on all models [46]. The bootstrap sampling for each regression tree generation and the random selection of features considered for partitioning at each node reduces the correlation between the generated regression trees, thus averaging their prediction responses to reduce the variance of the error is expected [45]. RF carries all the advantages of a decision tree with the added effectiveness of using several models [47,48]. RF is appropriate for modeling the non-linear effect of variables. The fact that RFR is nonparametric, and thus data do not need to come from a specific distribution, is not affected by multicollinearity and works well with many predictors [46,48,49]. According to Nolan et al. [50], RF is relatively robust to outliers. It can overcome ANN's black-box limitations by assessing the explanatory variables' relative importance and selecting the most important features.

### 2.4.3. Gradient Boost (GB)

Friedman [44] created the GB algorithm, one of the common ensemble boosting algorithms in which weak learners are trained iteratively and stage by stage to find a model that decreases prediction bias and variance. Like other boosting techniques, it assembles the model stage-wise and generalizes the model by optimizing a suitable cost function. In gradient boosting techniques, decision trees are utilized as weak learners. Each predictor in gradient boosting corrects the error of its predecessor. In contrast to AB, the training instance weights are not lowered; instead, each predictor is trained using

preceding residual errors as labels. In the GB algorithm, incorrectly classified cases for a step are given increased weight during the next step.

### 2.4.4. Support Vector Machine (SVM)

SVM is a vector-based statistical learning technique with proven predictive capability [12]. It is a machine learning tool for classification and regression. SVM is implemented using a kernel function, a non-linear mapping function. The kernel function and a hyperplane linearly separate and transform the input data points into a high-dimensional space. As a result, the choice of kernel function significantly impacts model accuracy [51]. The best way to choose the kernel function is to change the hyperplanes and reduce the errors associated with them iteratively [12].

### 2.4.5. Extreme Gradient Boost (XGB)

XGB, proposed by Chen and Guestrin [52], is a scalable artificial intelligence algorithm for tree boosting. XGB is one of the implementations of the technique of GB, which is one of the best performing algorithms for supervised learning. XGB can be used to solve problems involving regression and classification. XGB improves prediction performance by modifying the objective function of the GB algorithm to reduce model bias.

### 2.4.6. ANN

The ANN model is a non-linear regression model that uses a set of feed-forward neural networks to conduct an input–output mapping. It comprises three layers: an input layer, one or more hidden levels of computation nodes, and a computation node output layer. The highly linked framework of ANN models is recognized for transmitting information from the input layer through weighted connections and functional nodes known as transfer functions. These transfer functions make non-linear data mapping to high-dimensional hyperplanes easier, allowing for the separation of data patterns and the formulation of model output. ANN has been found to be fast and efficient and used to handle a wide range of problems [37]. We used one of the most common ANN structures utilized by many researchers: MLP architecture. MLP architecture has the advantage of being easy to use. It can approximate any relationship between input and output through the typical three layers [53], the input, hidden, and output layers. In this study, the most common transfer function, the sigmoid transfer function, was used in the hidden layer, while a linear activation function was used at the input and output layers.

### 2.5. Model Performance

For a comprehensive examination of model performance, the study employed statistical metrics, such as the determination coefficient ($R^2$), root mean square error (*RMSE*) Nash–Sutcliff efficiency (*NSE*), and mean absolute relative error (*MARE*) [54].

$$R^2 = 1 - \frac{\sum_{i=1}^{n}(Y - y_i)^2}{\sqrt{\sum_{i=1}^{n}(\overline{y} - y_i)^2}} \tag{1}$$

$$RMSE = \sqrt{\frac{\sum_{i=1}^{n}(y_i - \overline{y})^2}{n}} \tag{2}$$

$$MARE = \frac{1}{n}\sum_{i=1}^{n}\left|\frac{y_i - \overline{y}}{y_i}\right| \tag{3}$$

$$NSE = 1 - \frac{\sum_{i=1}^{n}(y_i - \overline{y})^2}{\sum_{i=1}^{n}\left(\overline{y} - \frac{\sum_{i=1}^{n} y_i}{n}\right)^2} \tag{4}$$

where ($Y$, $y_i$, $\overline{y}$, $n$) are mean true value, truth value, predicted value, and the number of data, respectively.

## 3. Results

### 3.1. Feature Selection for Data Input

Various visual spectral bands and their ratios are widely used to quantify water quality parameters [20]. This study originally proposed 78 spectral bands and their ratios, as in Table S1 of the Supplementary Material. Using the base algorithms of each of the selected ML models and the RFECV technique, the features were reduced for the three water quality parameters to speed up the learning algorithm, improve predicted accuracy, and make the learning results more intelligible. In the process of feature dimension reduction, the characteristic bands and band combinations for the three water quality parameters were found. The results of the feature selection analysis are presented in Table 2. The number of selected features for each model except ANN was greatly reduced (from 87 to a number ranging from 10 to 20) since many features were found to have little effect on these models' performances. ANN used all features as input due to the difficulty of automatic feature selection. Next, RF used a greater number of features; 20, 18, and 15 for Chl-a, TDS, and TUR, respectively. The selected features for each water quality parameter were not the same for different ML models. This implies that the correlation of the water quality parameter with the remote sensing image is not only affected by the relationship of the water quality parameter with features extracted from the image but also the ML models used. Chl-a were, for example, correlated with a few individual bands of the Landsat 8 OLI image in most of the models except in the SVM model, in which it was correlated with six of the individual bands. The selected features are also listed with their influence, from most important to least important, in Table 2. For example, the first most influential band is B1 for XGB algorithm to predict Chl-a. The second and third most influential features were found to be B2 and B10.

### 3.2. Comparision of ML Models' Performances Metrics

The performance metrics using Equations (1)–(4) for each model are shown in Table 3 for Chl-a, TDS, and TUR. Although XGB, RF, and AB had a very close value of $R^2$, their NSE values vary slightly. Figures 3–5 are scatter plots of the measured and predicted water quality parameters of Chl-a, TDS, and TUR, respectively, for the test dataset. The performance metrics result of the ML models for Chl-a with the test dataset showed good performance except for ANN (Figure 3). Although XGB, RF, and AB had a very close value of R2, their NSE values vary slightly. XGB performed best at capturing the relationship of Chl-a and the remote sensing image, with an $R^2$ of 0.78, NSE of 0.78, MARE of 0.082, and RMSE of 9.79 µg/L. GB was the second-best performing algorithm, with an $R^2$ of 0.77, NSE of 0.78, MARE of 0.091, and RMSE of 5.85 mg/L. RF's performance was third best with similar performance metrics, followed by AB and SVM. The lowest performing ML algorithm for predicting Chl-a was ANN, with an $R^2$ of 0.55, NSE of 0.056, MARE of 0.124, and RMSE of 7.14 µg/L.

**Table 2.** Selected features for the different algorithms.

| Water Quality Parameter | Models | Number of Features Selected | Selected Features |
|---|---|---|---|
| Chl-a | AB | 12 | B6, B7, B11, CI, GNDVI_4, TWI_2, Norm R, NDSI, SRSWIR1/NIR, ABI, B4/B3, (B3 + B4 + B5)/3 |
| | RF | 20 | B1, B3, B4, B10, CI, GDVI, EVI, TWI_2, GARI, NormR, NDSI, SRSWIR1/NIR, SRSWIR2/NIR, ABI, FAI, (B4 + B2)/2, (B4 + B5)/2, (B2 + B3 + B5)/3, (B3 + B4 + B5)/3, (B5 − B4)/(B2 + B3) |
| | GB | 10 | B2, B3, B6, B11, GNDVI_4, TWI_1, GARI, ABI, (B4 + B5)/2, (B2 − B4)/B3 |
| | SVM | 15 | B2, B3, B4, B6, B7, B11, CI, GNDVI_1, GNDVI_4, PPR, MSRNir/Red, RGR, (B4 + B5)/2, (B3 + B4 + B5)/3, (B2 − B4)/B3 |
| | XGB | 12 | B1, B2, B10, CI, GNDVI_3, GNDVI_5, TWI_1, TWI_2, Laterite, H, IF, SRSWIR1/NIR, SRSWIR2/NIR, FAI |
| | ANN | 87 | See Table S1 |
| TDS | AB | 15 | B3, B4, B7, B10, B11, TWI_1, BNDVI, GARI, Laterite, mCRIG, NDSI, B4/B3, (B3 + B5)/2, (B2 + B5)/2, (B4 + B5)/2 |
| | RF | 18 | B1, B3, B4, B6, B10, B11, GNDVI_3, GNDVI_4, MNDWI_2, TWI_2, Gossan, I, MVI, AWEInsh, (B4 + B3)/2, (B4 + B2)/2, (B3 + B2)/2, (B2 + B3 + B4)/3 |
| | GB | 10 | B2, B3, CI, TWI_1, GARI, PPR, Laterite, ABI, (B4 + B3)/2, (B2 + B3 + B5)/3 |
| | SVM | 13 | B10, B11, TWI_1, BNDVI, GARI, Laterite, mCRIG, ABI, FAI, (B4 + B2)/2, (B2 + B3 + B5)/3, (B3 + B4 + B5)/3, (B2 − B4)/B3 |
| | XGB | 10 | B2, B3, B7, B10, B11, CI, GNDVI_4, GNDVI_6, TWI_2, Gossan |
| | ANN | 87 | See Table S1 |
| Turbidity | AB | 15 | B3, B4, B7, B10, B11, TWI_1, BNDVI, GARI, Laterite, mCRIG, NDSI, B4/B3, (B3 + B5)/2, (B2 + B5)/2, (B4 + B5)/2 |
| | RF | 15 | B3, B4, B10, CI, GNDVI_3, TWI_1, TWI_2, Gossan, GARI, PVR, I, MVI, IF, SR550/670, SRSWIR1/NIR, SRSWIR2/NIR, RGR, FAI, (B4 + B3)/2, (B2 + B3 + B4)/3 |
| | GB | 10 | B2, B3, CI, TWI_1, GARI, PPR, Laterite, ABI, (B4 + B3)/2, (B2 + B3 + B5)/3 |
| | SVM | 13 | B10, B11, TWI_1, BNDVI, GARI, Laterite, mCRIG, ABI, FAI, (B4 + B2)/2, (B2 + B3 + B5)/3, (B3 + B4 + B5)/3, (B2 − B4)/B3 |
| | XGB | 10 | B2, B3, B7, B10, B11, CI, GNDVI_4, GNDVI_6, TWI_2, Gossan |
| | ANN | 87 | See Table S1 |

**Table 3.** Optical water quality parameters model prediction performance measure results using six ML algorithms; $R^2$, MARE, RMSE, and NSE are shown for Chl-a, TDS, and TUR.

| Water Quality Parameter | Algorithm | $R^2$ (TRST) | $R^2$ (STBPT) | MARE (TRST) | MARE (STBPT) | RMSE (TRST) | RMSE (STBPT) | NSE (TRST) | NSE (STBPT) |
|---|---|---|---|---|---|---|---|---|---|
| Chl-a (µg/L) | ANN | 0.55 | | 0.124 | | 7.14 | | 0.56 | |
| | XGB | 0.78 | | 0.082 | | 9.79 | | 0.78 | |
| | SVM | 0.67 | | 0.120 | | 8.27 | | 0.65 | |
| | GB | 0.77 | | 0.091 | | 5.85 | | 0.78 | |
| | AB | 0.74 | | 0.095 | | 7.32 | | 0.71 | |
| | RF | 0.77 | | 0.093 | | 6.81 | | 0.77 | |
| TDS (mg/L) | ANN | 0.60 | | 0.133 | | 17.04 | | 0.58 | |
| | XGB | 0.78 | | 0.085 | | 12.51 | | 0.78 | |
| | SVM | 0.61 | | 0.112 | | 16.86 | | 0.62 | |
| | GB | 0.79 | | 0.096 | | 12.40 | | 0.78 | |
| | AB | 0.77 | | 0.095 | | 12.99 | | 0.77 | |
| | RF | 0.79 | | 0.082 | | 12.30 | | 0.80 | |
| TUR (NTU) | ANN | 0.22 | 0.60 | 0.33 | 0.132 | 30.1 | 10.97 | 0.34 | 0.61 |
| | XGB | 0.53 | 0.79 | 0.20 | 0.076 | 24.5 | 8.05 | 0.55 | 0.80 |
| | SVM | 0.23 | 0.64 | 0.24 | 0.122 | 15.6 | 10.17 | 0.35 | 0.64 |
| | GB | 0.45 | 0.77 | 0.22 | 0.085 | 21.5 | 8.26 | 0.46 | 0.77 |
| | AB | 0.44 | 0.74 | 0.21 | 0.092 | 13.5 | 8.60 | 0.45 | 0.75 |
| | RF | 0.48 | 0.80 | 0.26 | 0.072 | 18.4 | 7.82 | 0.48 | 0.81 |

TRST refers to traditional random sampling techniques, and STBPT refers to spatiotemporal block partition techniques.

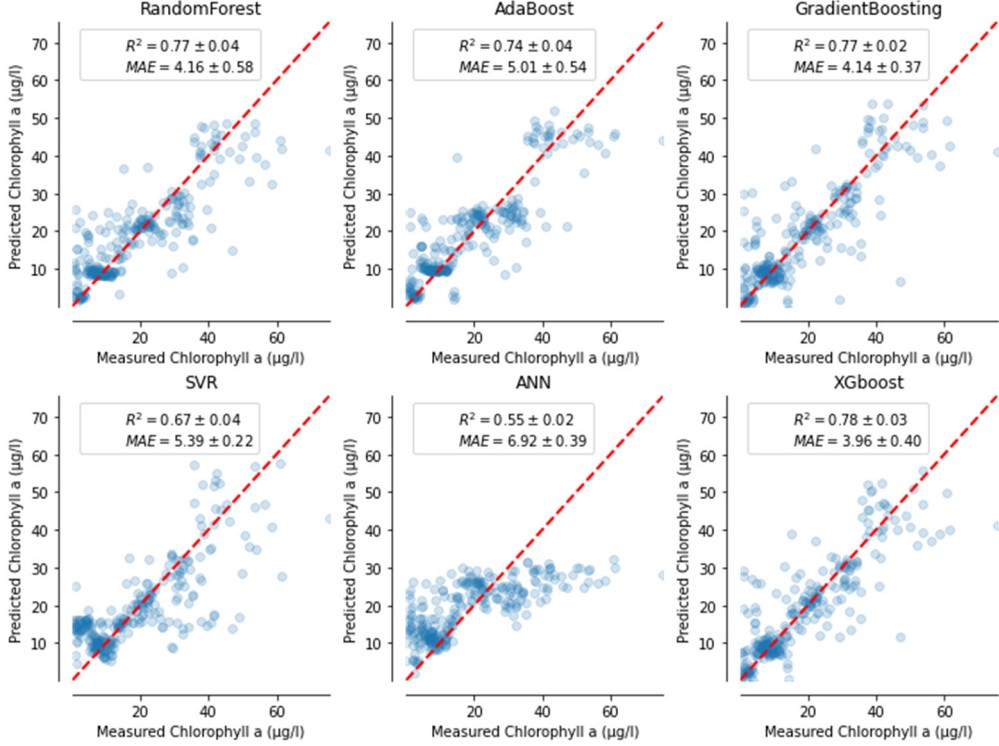

**Figure 3.** Model performance using a test set for Chl-a (µg/L) using six ML methods.

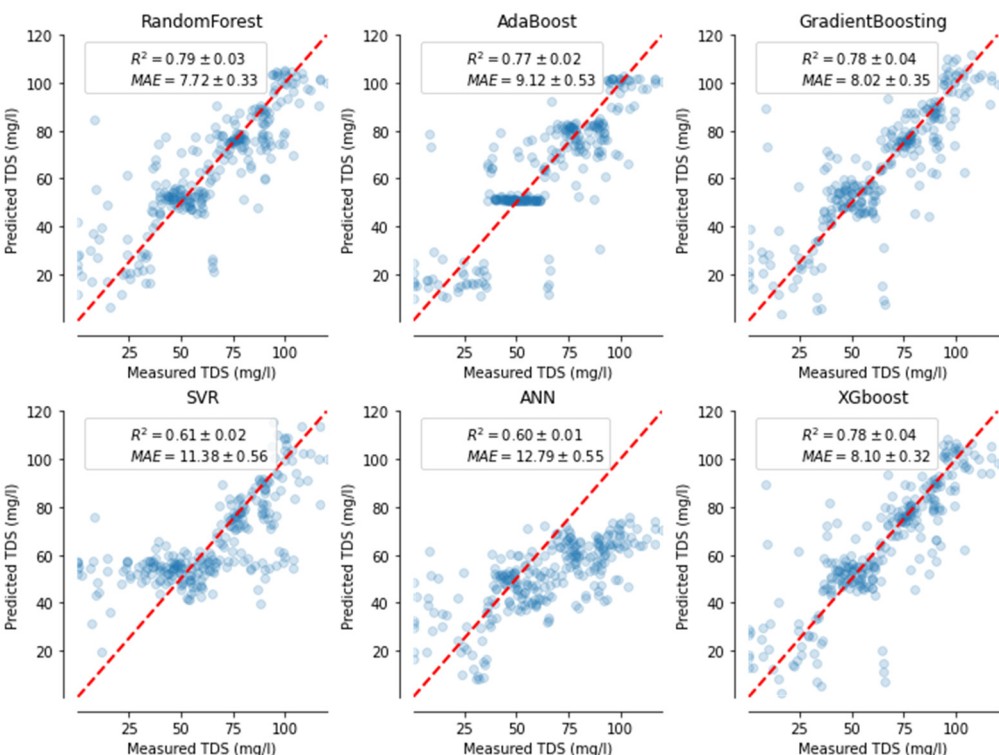

**Figure 4.** Model performance using a test set for TDS (mg/L) using six ML methods.

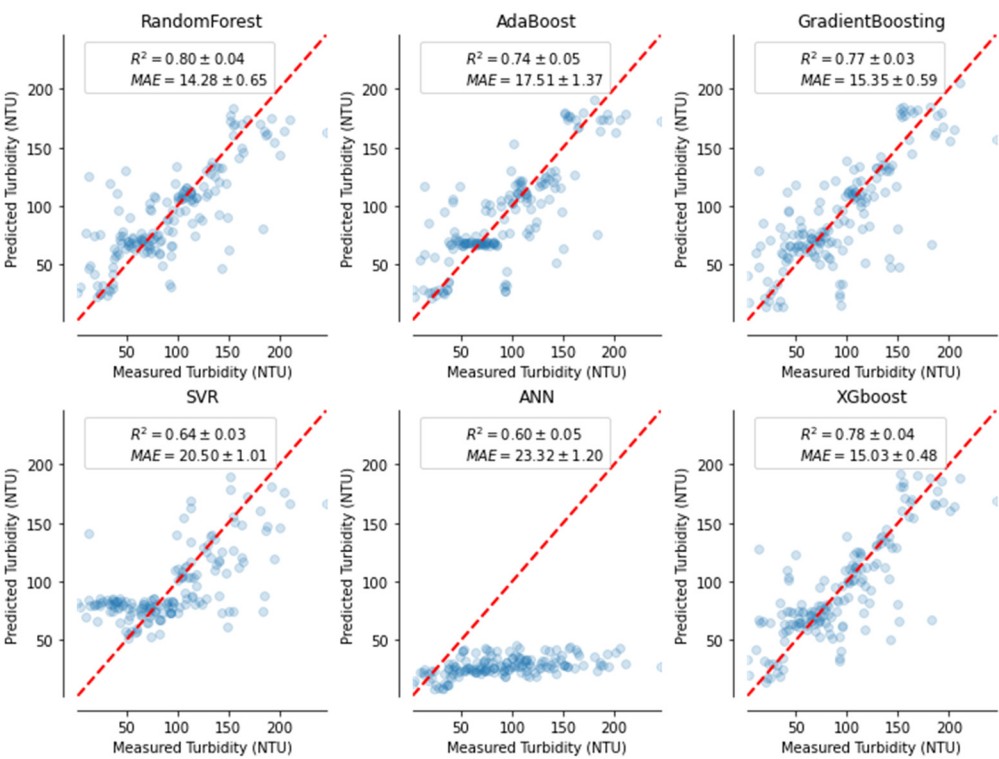

**Figure 5.** Model performance using test set for turbidity (NTU) using six ML methods.

The performances of the ML models for TDS were similar to Chl-a (Table 3, Figures 3 and 4). Although ANN was more accurate at capturing the relationship of TDS and remote sensing images than Chl-a, it was still the poorest performing model, with an $R^2$ of 0.60, NSE of 0.58, MARE of 0.133 and RMSE of 17.04 mg/L. ANN has the largest error (MARE of 0.133 and RMSE of 17.04). RF was the best performing method, with an $R^2$

of 0.79, NSE of 0.80, MARE of 0.082, and RMSE of 12.30 mg/L. XGB also performed well with an $R^2$ of 0.78, NSE of 0.78, MARE of 0.085, and RMSE of 12.51 mg/L. GB, AB, and SVM were third, fourth, and fifth best, respectively, with slight variations in performance metrics. Although RF and XGB could be used for TDS retrieval, we selected RF for further prediction analysis.

TUR data used for training and testing was the smallest (only two months). At first, the sampling technique employed for data splitting into training and test set using traditional random sampling techniques (TRST) had resulted in very poor performances of ML models (Table 3). We then improved the models' performances by applying a different sampling technique: the spatiotemporal block partition technique (SPBPT) sampling technique. The result was shown under the SPBPT column for each metric in Table 3 and Figure 5. The best performing algorithm using this technique was RF, with an $R^2$ of 0.80, NSE of 0.80, MARE of 0.072, and RMSE of 7.82 NTU. The second-best performing algorithm was XGB, with an $R^2$ of 0.79, NSE of 0.80, MARE of 0.076, and RMSE of 8.05 NTU. GB was third best, with an $R^2$ of 0.77, NSE of 0.77, MARE of 0.085, and RMSE of 8.26 NTU. AB, SVM, and ANN all performed at lower accuracy in capturing the relationship between TUR and the remote sensing image, with ANN being the worst. RF and AB have the best fitting accuracy ($R^2$ and NSE), and XGB has the lowest error (MARE). Thus, RF, AB or XGB could be used for TUR prediction.

## 4. Discussions

In this section, we first discussed the observed data in Table 1. Then, to validate the ML models' applicability, we predicted monthly water quality parameters in 2017 and 2020. For Cla-a, 2020 was chosen, and for TDS and TUR, 2017 was selected. These years were selected to compare our predictions with previous studies by Dersseh et al. [15], Worqlul et al. [40], and Mucheye et al. [41]. In addition, we compared the prediction with the observed data in Table 1. The comparison is to check how the models capture the spatial and seasonal trends and order of magnitude.

### 4.1. Chl-a Distribution of Lake Tana

The mean Chl-a concentration of the lake from the observed data was below 20.6 μg/L in all years (Table 1). The most recent Chl-a concentrations were 7 μg/L in October, 2021, and 9 μg/L in April, 2022. These observed Chl-a were similar to measurements by Dersseh et al. [15] in order of magnitude; however, the observed Chl-a concentration in December, 2016 (17 μg/L) and March, 2017 (21 μg/L) were different from the measurements by Dersseh et al. [15] in 2019 and 2020.

This study used the best-trained and validated ML models XGB to retrieve Chl-a for the year 2020 from Landsat 8 OLI images. Based on the feature selection, the bands and band combinations used for the prediction were B1, B2, B10, CI, GNDVI_3, GNDVI_5, TWI_1, TWI_2, Laterite, H, IF, SRSWIR1/NIR, SRSWIR2/NIR, and FAI (Table 2). Figure 6 shows the monthly retrieved Chl-a distribution in Lake Tana. The results showed a high spatiotemporal variation of Chl-a concentration over Lake Tana in certain months. From March to June, the highest concentration predicted was in the western and central parts of the lake, possibly due to the effects of wind [10]. From observed data by Dersseh et al. [15], a similar spatial pattern of higher concentration in the west and lower concentration in the east was observed. From July to September, the predicted Chl-a concentration looks spatially uniform. This could have been due to the increased mixing of lake water during the rainy season, as suggested by Wondie et al. [42].

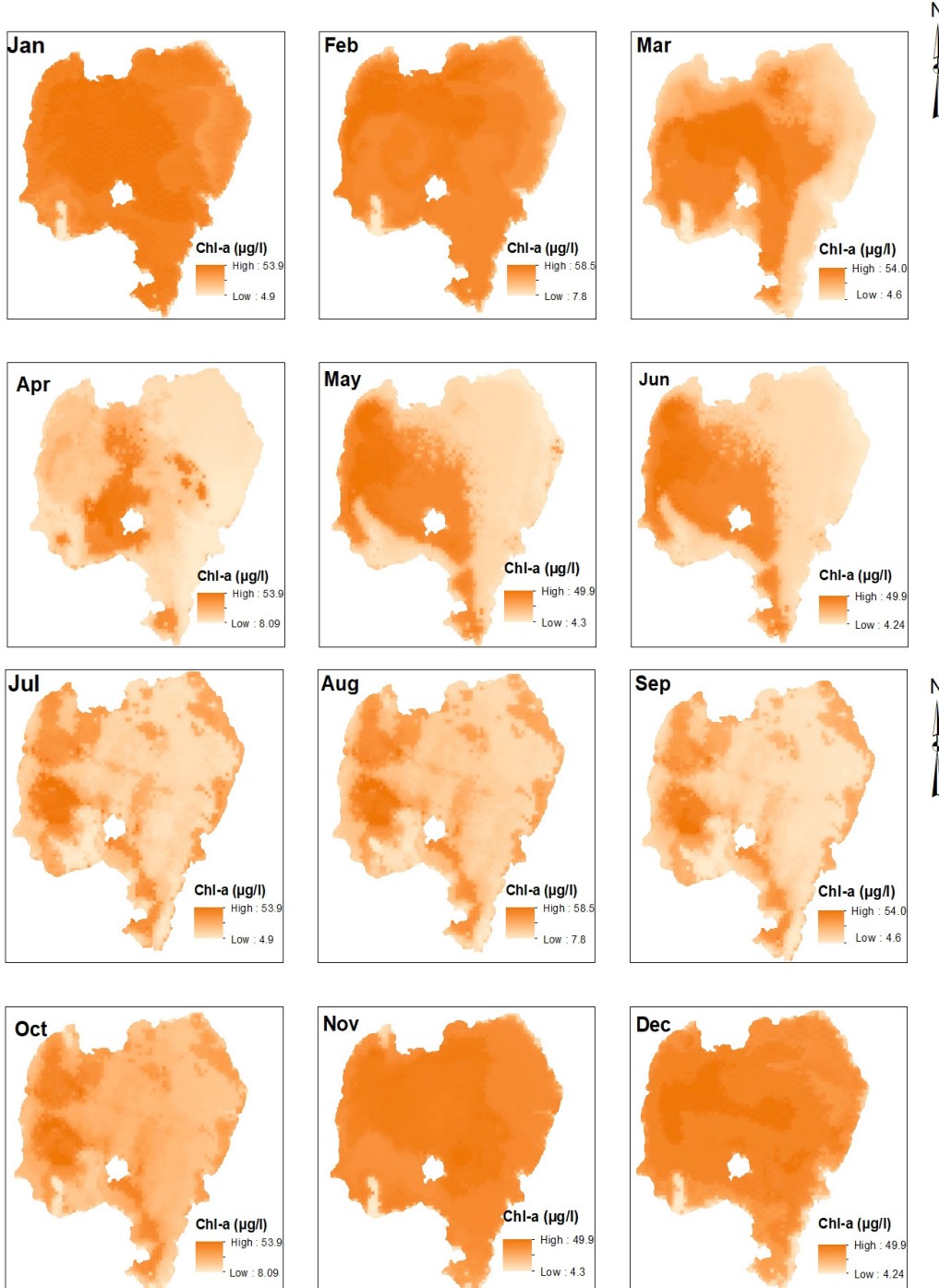

**Figure 6.** Monthly Chl-a (µg/L) spatial variation predicted using the XGB regression algorithm for the year 2020.

Temporally, for the year 2020, the maximum predicted Chl-a concentration was in February (58.5 µg/L), and the minimum predicted Chl-a concentration (4.2 µg/L) was in June (Figure 6). The range in the predicted Chl-a was smaller as opposed to the observed high range (Table 1). The range was smaller in the predicted Cla-a concentrations across the months in Lake Tana. Hence, it can be noticed that the XGB model under-predicted the maximum observed Chl-a concentration (191.6 µg/L) in December and March and over-predicted the minimum observed Chl-a concentration (0.05 µg/L) in August. These could be due to the outlier removal technique we applied to the model during training to improve the model's accuracy. Predictions by Mucheye et al. [41] from sentinel indicated that the

maximum Chl-a concentration was 40.0 μg/L by the end of August, and the minimum Chl-a concentration was 4.41 μg/L in June. The highest predicted spatial coverage of Chl-a concentration from observation was after the rainy season and not during the main rainy period [42]. Therefore, our predictions of Chl-a concentration appear to be more accurate than those of Mucheye et al. [41], who predicted the highest Chl-a coverage during the rainy period.

In general, Lake Tana underwent major water quality changes over time. Accurate predictions help decisionmakers to understand the full extent of water quality degradation in the lake. For example, since 2003, the average spatial Chl-a concentration has increased from 4.8 μg/L, according to Wondie et al. [42], to 32.7 μg/L in 2020. Thus, there has been an eight-fold increase in Chl-a within just two decades.

### 4.2. TDS Distribution of Lake Tana

The average TDS of the lake in Table 1 were just below 100 mg/L in 2016 and 2017 but lower (59 mg/L and 68 mg/L, respectively) when measured in October, 2021 and April, 2022. Previous studies did not attempt to predict the spatiotemporal distribution of TDS in the Lake. Of the six algorithms considered in this study, RF performed best at predicting TDS. This algorithm predicted the monthly TDS content across Lake Tana for 2017, as shown in Figure 7.

It used the following features, B1, B3, B4, B6, B10, B11, GNDVI_3, GNDVI_4, MNDWI_2, TWI_2, Gossan, I, MVI, AWEInsh, (B4 + B3)/2, (B4 + B2)/2, (B3 + B2)/2, (B2 + B3 + B4)/3, to perform the prediction (Table 2). Spatial monthly variations of TDS concentrations were highest in the main rainy season of July, August, and September. The predictions showed high concentrations of TDS in the green-colored areas, especially around the Gilgel Abay River in the southwest. These predicted spatial patterns seem to accurately reflect Gilgel Abay's role as a major source of sediment in the lake [15].

The predicted maximum TDS content was 87.5 mg/L in July, August, and September of 2017, around the major river tributaries of Gilgel Abay, Gumera, and Ribb rivers. The predicted minimum TDS content was 49.3 mg/L in the same months of July and August around the lake's center. RF prediction showed that the predicted TDS content was within the observed maximum TDS content, 113.3 mg/L, observed in August, and the observed minimum TDS content, 7.3 mg/L. The spatial variability of TDS along the season has no such great variation. This is because incoming solids from the rivers settle on the lake bottom during the rainy period and mix during the dry period as the lake is shallow, elevating the TDS concentration in the water column. The area of the lake coverage with the highest TDS concentration is observed in September and October, while the concentration of the area at river inlets is elevated in July and August. There is a delay compared to the rivers, and the same pattern from observation data were reported by Dersseh et al. [15].

Figure 7 shows that the monthly TDS content increased when entering the month of June in the rainy season, peaked in October, and then decreased in the following months. Therefore, the model's predictions are consistent with observed TDS content in Table 1. The observed maximum TDS content was 113.3 mg/L in December, 2016 and March, 2017, higher than the predicted maximum of 87.5 mg/L in July, August, and September, 2017. The observed minimum TDS content was 7.3 mg/L in April, 2022, less than the predicted minimum of 48.3 mg/L in July, 2017. Hence, the predicted TDS values were within the observed range but showed inconsistency with the months. RF fails to predict outside the range of observed values used for its training [55]. Compared to the observed TDS content, RF under-predicted maximum TDS by 22% and over-predicted minimum TDS by a factor of 5.

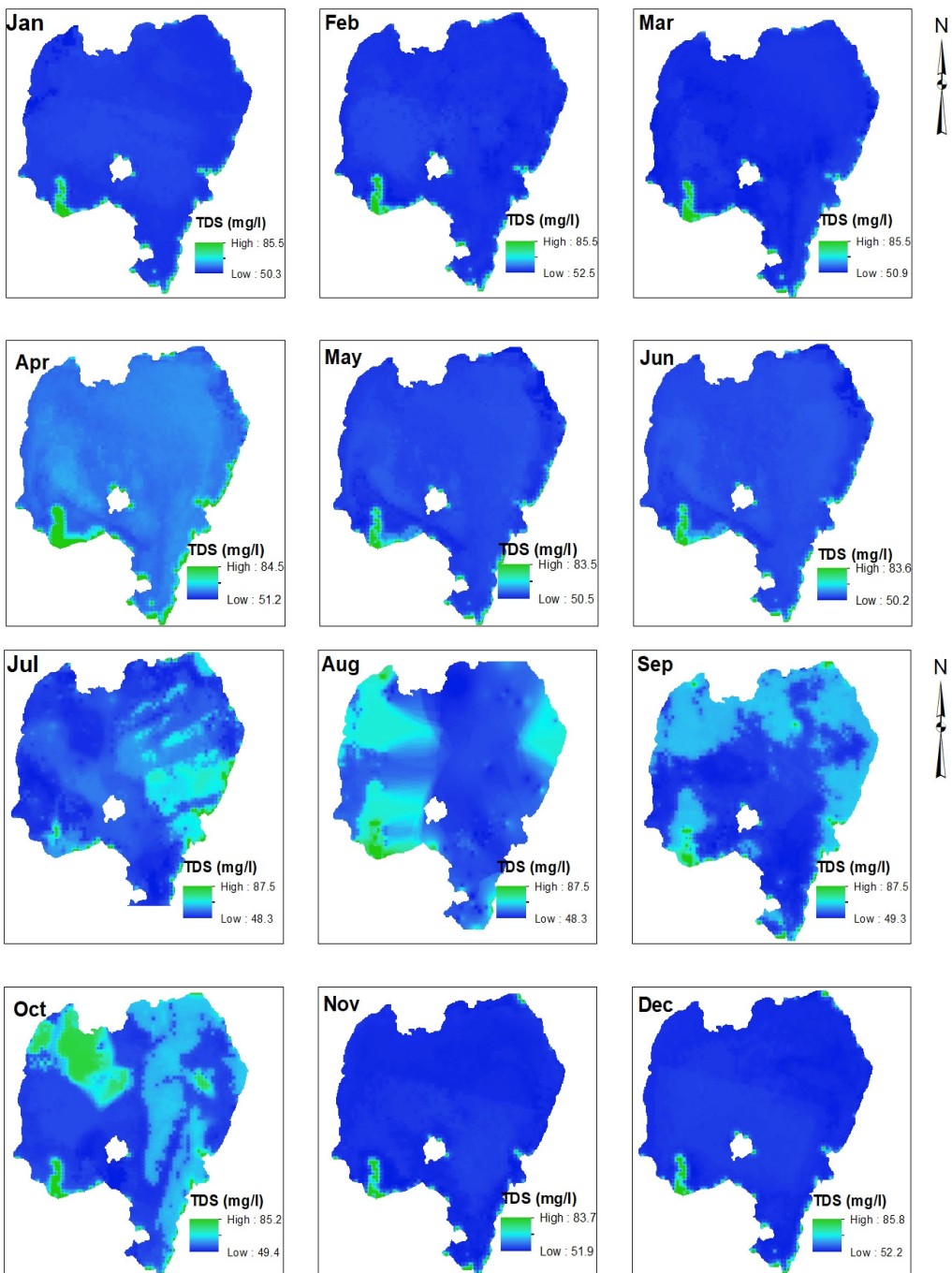

**Figure 7.** Monthly TDS (mg/L) spatial variation predicted by RF regression algorithm for 2017.

### 4.3. TUR Distribution of Lake Tana

The observed TUR was, on average, close to 40 NTU, and these mean values were within the range measured by Zelalem et al. [56]. Additionally, using the RF algorithm, TUR was predicted for the year 2017 on a monthly basis, using B3, B4, B10, CI, GNDVI_3, TWI_1, TWI_2, Gossan, GARI, PVR, I, MVI, IF, SR550/670, SRSWIR1/NIR, SRSWIR2/NIR, RGR, FAI, (B4 + B3)/2, (B2 + B3 + B4)/3 bands and band combinations, as shown in Figure 8. The maximum predicted monthly TUR was 145.2 nephelometric turbidity units (NTU) in July and August (in the main rainy season), and the minimum predicted monthly TUR was 39.5 NTU in November.

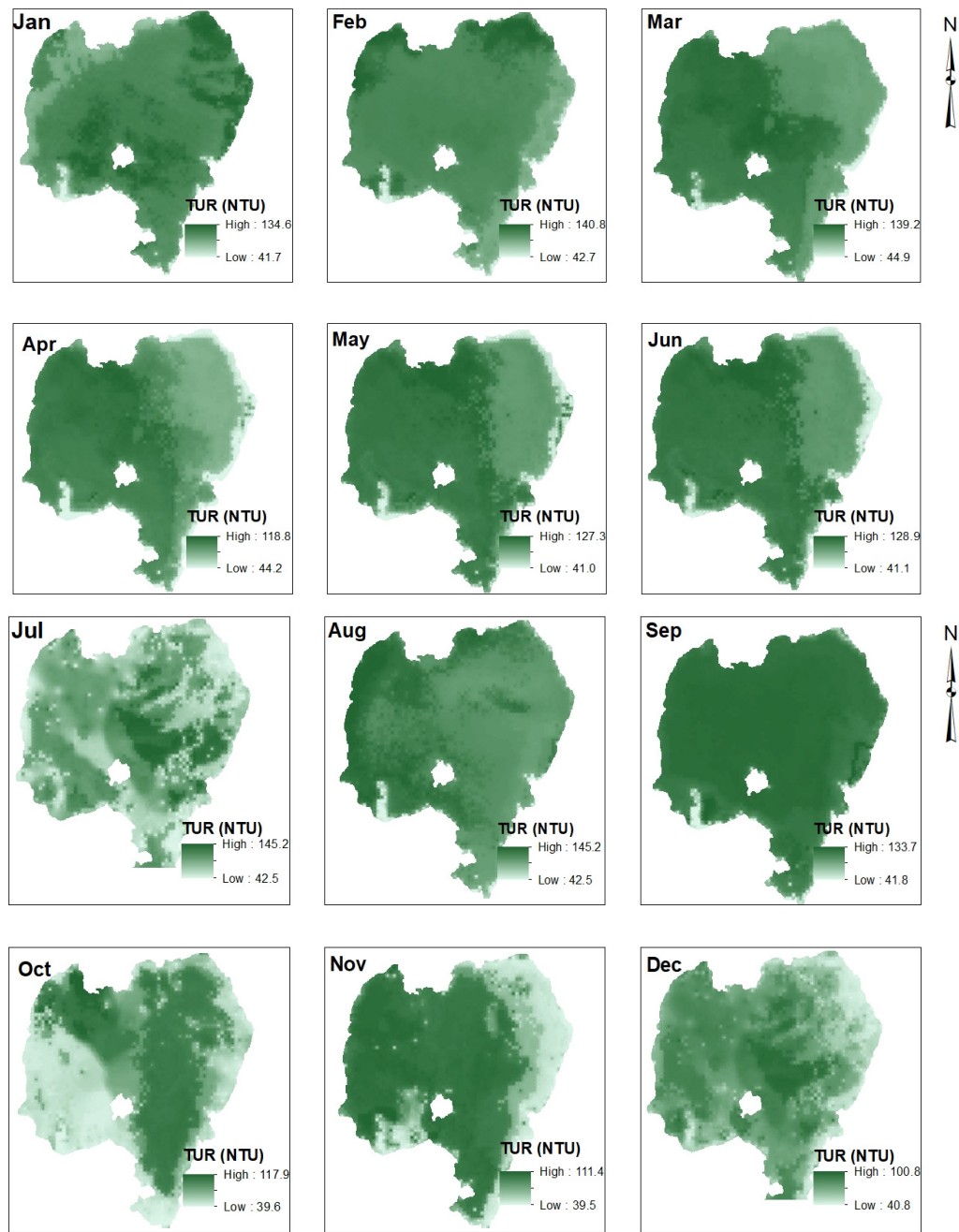

**Figure 8.** Monthly TUR (NTU) spatial variation predicted by RF regression algorithm for 2017.

The spatiotemporal distribution pattern of predicted TUR in this study was similar to that shown in Worqlul et al. [40]; however, the difference in the magnitudes of the two studies was very large, indicating the need for further modeling with a more temporally and spatially extensive dataset. The average annual TUR of the lake reported in [40] was 348 NTU, greater than 10 times the observed average annual TUR listed in Table 1. This estimate by Worqlul et al. [40] was based on the relationship between in situ measurements of TUR (NTU) and reflectance of MODIS near-infrared channel (NIR) developed more than six years prior. This relationship may vary with time and might need to be modified. It is also worth noting that vegetation cover (such as water hyacinth) [6] and summer cloud cover could affect spectral characteristics and the algorithms' predictions. The impacts of environmental factors such as wind speed, vegetation cover offshore, and sediment nutrient released on the spatiotemporal distribution of Chl-a, TDS, and TUR also merit further investigation.

### 4.4. Models Performances and Their Limitations

Data sizes of 931 and 796 spanning over several months and years were used to develop the Chl-a and TDS retrieval model. For TUR, a smaller data size of 286 was collected for two months in two years. There was an apparent limitation of data size, frequency of data sampling and sampling resolution. Despite these challenges, the result indicated that the ML algorithms generally performed well for Chl-a, TDS, and TUR modeling. In the case of the Chl-a retrieval model building, XGB performed best. It did not only have the best fitting accuracy ($R^2$ and NSE) but also the lowest error rates (MARE and RMSE). In the case of TDS and TUR, RF performed well with slightly higher model fitting accuracy metrics $R^2$ and NSE. It also had the lowest error rates in both MARE and RMSE metrics.

Despite the overall good performances of the ML models, the presence of data imbalance and differences in the number of features used in the ML models has had the main impact on the accuracy of the models. Chl-a had relatively longer data, but it also had a larger data imbalance than TDS and TUR due to measurements from offshore for some of the months. On the other hand, RF used more features than the other ML models and has the ability to extract more information from large features (Table 2). As proved in previous studies, XGB performed well in the presence of high data imbalance [53,57] situations, while RF worked well in high-dimensional data [47,48]. Thus, the ML model performance result could have described the differences in the data imbalance and dimensionality used by these ML models. Moreover, in the case of TUR retrieval model building, again, RF performed best. That could be due to its stability with the variation of sampling size [49].

To further evaluate the performance of the best models in this study, we retrieved Chl-a using XGB and TDS and TUR using RF from remote sensing, as presented in Figures 6–8. The ML models in all cases did not fully capture the temporal variability and spatial distributions of the water quality parameters. This could likely be due to ML models' large uncertainties associated with their unique structures, hyper-parameter adjustment requirements, and data inputs [6,7]. The data imbalance and spatial heterogeneity that is in embedded in our spatiotemporal data may have introduced bias into our modeling. Additionally, ML was known to be affected by sampling frequencies [8,58,59]. Our data's monthly time step and 5 km spatial resolution might have a limited representation of the underlying water quality dynamics and its relation with remote sensing images. Collecting enough data at high temporal and spatial frequencies can improve these predictive models.

In the case of TUR modeling, we initially applied the same training and test data split technique (i.e., spatiotemporal data partitioning using traditional random sampling techniques (TRST)) that we applied for Chl-a and TDS modeling. However, five times more data were collected for Chl-a and TDS than for TUR. All TUR data may have been used for training in certain locations, with none left for testing using the TRST method. This could lead to loss of information or data leakage [39]. Additionally, significant variations in the sizes of collected data samples may have caused biased learning, which can result in the poor performance of predictive models, as noted in Weiss and Provost, 2003 [58]. To improve the performances of the algorithms for TUR modeling, we subsequently employed an STBPT or stratified sampling to address the issues of imbalanced data and the limitations of the traditional random sampling technique with respect to spatiotemporal data. Following the change in the sampling technique, it was managed to improve the models' performances for TUR.

Overall, the ensemble-type algorithms (RF, XGB, AB, and GB) produced higher prediction accuracy than SVM and ANN in all three water quality parameters. They have proven to be robust algorithms with better generalization abilities and are less affected by overfitting than the ANN and SVM algorithms. The strength of the ensemble methods could also relate to their advantage in handling imbalanced datasets, as demonstrated by Leevy et al. [57]. Of the ensemble methods, the RF and XGB algorithms performed best, and thus proved to be the most robust algorithms, as indicated in prior studies [49,50]. The AB algorithm also performed well, but, as others noted, the efficiency of the technique is highly affected by outliers and easily overwhelmed by noisy data [59].

The inefficiencies of this model could cause the poor performance of ANN. A trial-and-error method to determine the ANN structure does not fundamentally promote the further development of the model [60], making it difficult to improve the model's performance. It could also be that the optimization of model parameters in a neural network is unstable, such that the model's accuracy is remarkably affected by non-linear disturbances [13]. Furthermore, the "black box" nature of neural networks means that the relationships between the response and predictor variables may be unclear [61]. In addition, the training process in ANN takes longer; overfitting problems may occur if there are too many layers, while prediction accuracy may be affected if there are not enough layers [62]. ANN can be obstructed if the training data are imbalanced and when all initial parameter weights have the same value. Although ANN models are the most broadly used ML models, their predictive power is weakened if they are used with a small dataset and the testing data are outside the range of the training data [9].

Moreover, our data were not considered continuous monthly data. Chl-a and TDS are collected for more than 6 months and can capture the seasonal variation before and after rainy and dry periods. The major limitation of the data was for TUR, which was collected only for two months. Due to the larger size of the lake, the data collection took more than two weeks, which did not exactly coincide with the date of the satellite overpass of the lake. In addition, our approach did not consider the effect of wind and precipitation on the spatiotemporal distribution of the water quality parameters and hence the remote sensing images. With such data limitation, we studied the ML models known to perform well in similar situations. Thus, despite some drawbacks, with limited data, our effort has shown these models can do well in tropical Ethiopia. Additionally, our findings have shown that not all ML models work well, but RF and XGB worked well for Chl-a and TDS, while only RF worked well for TUR.

*4.5. Significance of ML in Lake Water Quality Management*

The lake water quality has degraded slowly with time [15]. The explanation for this is that the amount of cultivated land in the Lake Tana area has increased by 20% and covers 68% of the basin in the last 30 years [63]. The soil erosion from these areas ranges from 5 tons per hectare per year to 50 tons per hectare per year, indicating a doubling of the sediment transport to the lake from 1980s to 2020s [64]. With the expansion of irrigated land from 540 $km^2$ in 1980s to 1200 $km^2$ in 2020s, agrochemical uses are expanded. With further water resource development and watershed management in the basin, the lake needs special strategy in monitoring the water quality.

Though traditional environmental monitoring methods are widely applied by different environmental agencies in the globe and are still useful, this is not even happening by responsible agencies in Ethiopia. Most of the water quality data for the lake have been made available by researchers. Conventional models have also been applied to carry out predictions in the absence of in situ data [65]. However, their temporal and spatial data demand limited calibration and reliable prediction. This research showed that data-driven models based on machine learning can efficiently solve more complex nonlinear problems. The data were obtained from researchers with some strategic in situ observation. This implies that agencies can plan cost effective in situ data collection and combine these data with remote sensing information to close the data gaps.

In this study, we showed that remote sensing information can meet the needs of data input and large-scale water quality monitoring, and can also be used to reveal the success of watershed management in improving quality. In Lake Tana basin, there are currently watershed management practices in the upland, rehabilitating wetlands by creating buffering zones and reforesting degraded land through the green legacy program since 2019, which could lead to the reduction of pollutant load to the lake. The prediction with ML can support finding the most effective best management practices in the area. In future work, with some additional in situ measurements, the impact of different best management practices could be studied with ML models identified in this study. While

this study does not exhaustively explore all ML models, the relationship established with remote sensing variables would provide insight into the factors affecting each water quality variable. These models have also the potential to be applied to other lakes in the same climate such as Victoria Lake in the east Africa region.

Finally, with future deployed real-time monitoring sensors and satellite data, there is a potential to forecast water quality and learn from natural processes in the region, as well as assess anthropogenic impacts on the lake's ecosystem.

## 5. Conclusions

This study applied six ML algorithms (AB, RF, GB, SVM, XGB, and ANN) to build retrieval models for three optical water quality parameters (Chl-a, TUR, and TDS) in an area where data are scarce and standard monitoring is very expensive (Lake Tana). It was found possible to develop reasonably accurate ML models that could be used to monitor Chl-a, TUR, and TDS in the absence of high-resolution field monitoring techniques. Monthly water quality maps were also retrieved using the best performing ML models. Despite the limitation of data for ML model training, the result of the study suggests that certain ensemble ML methods with satellite data have a staggering promising potential for regular water quality monitoring over large complex inland tropical lakes and other water bodies such as reservoirs. At this level, we recommend that the XGB algorithm (or the AB) could be used to retrieve Chl-a, and RF algorithm (or the XGB algorithms) could be used to retrieve TDS and TUR from Landsat 8 OLI imagery quickly and with reasonable accuracy. Such quickly produced and relatively accurate water quality maps (Chl-a, TDS, and TUR) could be used to identify sources of pollution, the water quality status of water bodies, the water quality trend, and the factors affecting their distribution in the water body. Subsequently, the retrieved maps could be used by different research institutes and policymakers to develop management and policy scenarios to protect the lake's water resources.

In the future, efforts would focus on improving the accuracy of the estimation of the overall water quality status of the lake, developing new methods to fit, retrieve, and monitor more factors that can represent the water quality status of the Lake and other water bodies in the region. Simultaneously, long-term spatiotemporal data management should be planned and implemented so that the data could be used to develop more accurate retrieval ML algorithms. In that case, ML models, such as convolutional neural networks (CNN) and long short-term memory (LSTM), that would provide better accuracy with time series spatiotemporal data could also be evaluated.

**Supplementary Materials:** The following supporting information can be downloaded at: https://www.mdpi.com/article/10.3390/hydrology10050110/s1, Figure S1: AdaBoost number of estimator tuning for Chl-atitle; Figure S2: AdaBoost learning rate tuning for Chl-a; Figure S3: Random Forest maximum depth tuning for Chl-a; Figure S4: Random Forest number of estimator tuning for Chl-a; Figure S5: SVR c regularization tuning for Chl-a; Figure S6: SVR gamma tuning for Chl-a; Figure S7: XGBoost learning rate tuning for Chl-a; Figure S8: XGBoost maximum depth tuning for Chl-a; Figure S9: GradBoost number of estimator tuning result for Chl-a; Figure S10: XGBoost learning rate tuning for Chl-a; Figure S11: GradBoost number of estimator tuning for Chl-a; Figure S12: GradBoost maximum depth tuning for Chl-a; Figure S13: AdaBoost learning rate tuning for TDS; Figure S14: AdaBoost number of estimator tuning for TDS; Figure S15: GradBoost learning rate tuning for TDS; Figure S16: GradBoost maximum depth tuning for TDS; Figure S17: GradBoost number of estimator tuning for TDS; Figure S18: Random Forest maximum depth tuning for TDS; Figure S19: Random Forest number of estimator tuning for TDS; Figure S20: SVR c regularization tuning for TDS; Figure S21: SVR gamma tuning for TDS; Figure S22: XGBoost learning rate tuning for TDS; Figure S23: XGBoost maximum depth tuning for TDS; Figure S24: XGBoost number of estimator tuning for TDS; Figure S25: AdaBoost learning rate tuning for TUR; Figure S26: GradBoost number of estimator tuning for TUR; Figure S27: Random Forest number of estimator tuning for TUR; Figure S28: Random Forest maximum depth tuning for TUR; Figure S29: SVR c regularization tuning for TUR; Figure S30: SVRgamma tuning for TUR; Figure S31: XGBoost learning rate tuning for TUR; Figure S32: XGBoost maximum depth tuning for TUR; Figure S33: XGBoost number of estimator tun-

ing for TUR; Figure S34: GradBoost learning rate tuning for TUR; Figure S35: GradBoost maximum depth tuning for TUR; Figure S36: GradBoost number of estimator tuning for TUR; Table S1: Water indices as features derived from Landsat 8 OLI used in the study.

**Author Contributions:** Conceptualization, E.S.L.; Data curation, E.S.L.; Formal analysis, E.S.L.; Investigation, E.S.L.; Methodology, E.S.L.; Resources, E.S.L.; Supervision, S.A.T., F.A.Z., D.S. and T.E.; Validation, E.S.L.; Writing—original draft, E.S.L.; Writing—review and editing, E.S.L., S.A.T., F.A.Z., D.S., R.S. and T.E.; funding acquisition, E.S.L. and S.A.T. All authors have read and agreed to the published version of the manuscript.

**Funding:** This research was funded by "the International Development Research Center (IDRC)" and "Swedish International Development Cooperation Agency (SIDA)" through Artificial Intelligence for Development–AI4D Africa (http://africa.ai4d.ai) to collect data after 2020 and support the first author.

**Data Availability Statement:** Not applicable.

**Acknowledgments:** This research was made possible through initial support provided by the U.S. Agency for International Development under the Feed the Future Evaluation of the Relationship between Sustainably Intensified Production Systems and Farm Family Nutrition (SIPS-IN) project (AID-OAA-L-14-00006) with additional support from the Feed the Future Innovation Lab for Small Scale Irrigation (contract no. AID-OAA-A-13-0055) to collect data before 2020. We would like to acknowledge the assistance of Minyechel Gitaw, Aron Ateka Abiy Gebeyhu, and Work Abunu for their valuable help to collect the necessary data for this study.

**Conflicts of Interest:** The authors declare no conflict of interest.

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
