# Peer review of "Predicting Optical Water Quality Indicators from Remote Sensing Using Machine Learning Algorithms in Tropical Highlands of Ethiopia"

_hydrology, doi:10.3390/hydrology10050110_

Round 1

Reviewer 1 Report

(1) As a technical paper, for readers to quickly catch your contribution, it would be better to highlight major difficulties and challenges, and your original achievements to overcome them, in a clearer way in abstract and introduction.

(2) In the section 2.2, it is suggested to provide more detailed sample data processing methods, such as the processing of outliers.

(3) In generally, the distribution ratio of model training data and verification data is 30% to 70%. Why is 20% data used for model training and 80% for model verification in this study? I think it is more convincing for you to verify it with data.

(4) Section 3.1 mentions that " For each water quality parameter, the selected features were not the same for different ML models", and does the difference in feature selection affect the fairness of comparison between models?

(5) In the Discussion section, an in-depth discussion should be given to
enhance the practical significance of the results and findings of the
current study, which will be more useful than just performing
statistical data collection and analysis.

This paper can be published only after the authors has fully addressed
all these comments.

Line 337, please pay attention to the writing rules. I strongly
recommend that authors double-check and correct them.

Author Response

Dear Reviewer,

We appreciate the time and effort that you and the reviewers have dedicated to providing your valuable feedback on our manuscript titled” Predicting Optical Water Quality Indicators from Remote Sensing using Machine Learning Algorithms in Tropical Highlands of Ethiopia. We have done our best to respond to each comment. We trust the manuscript will live up to your high expectations following rigorous edits. The manuscript has been modified, and all changes are with track changes and red-highlighted.  The point-by-point responses are provided below, with the revised text in red-color.

Reviewer 2 Report

The paper is well presented in terms of research background, methodology, results and discussion, and conclusions. Few corrections are required which are as follows:

1. At few places, spelling mistakes need to be corrected.

2. Add a common scale for Figures 6, 7, and 8. 

Quality of English is good and understandable. 

Author Response

(The authors gave the same response as above.)

Reviewer 3 Report

The manuscript is well written. The structure is very clear. The results support the study hypothesis very well.

N/A

Author Response

Dear Reviewer,

We appreciate the time and effort that you and the other reviewers have dedicated to providing your valuable feedback on our manuscript titled” Predicting Optical Water Quality Indicators from Remote Sensing using Machine Learning Algorithms in Tropical Highlands of Ethiopia. Thank you very much for your positive feedback on our work. We have done our best to respond to each comment from the other reviewers. We trust the manuscript will live up to your high expectations following rigorous edits. The manuscript has been modified, and all changes are with track changes and red-highlighted.  The point-by-point responses are provided below, with the revised text in red-color.